# Evolution and Applications of Recent Sensing Technology for Occupational Risk Assessment: A Rapid Review of the Literature

**DOI:** 10.3390/s22134841

**Published:** 2022-06-27

**Authors:** Giacomo Fanti, Andrea Spinazzè, Francesca Borghi, Sabrina Rovelli, Davide Campagnolo, Marta Keller, Andrea Borghi, Andrea Cattaneo, Emanuele Cauda, Domenico Maria Cavallo

**Affiliations:** 1Department of Science and High Technology, University of Insubria, 22100 Como, Italy; andrea.spinazze@uninsubria.it (A.S.); francesca.borghi@uninsubria.it (F.B.); sabrina.rovelli@uninsubria.it (S.R.); davide.campagnolo@uninsubria.it (D.C.); mkeller@uninsubria.it (M.K.); aborghi5@studenti.uninsubria.it (A.B.); andrea.cattaneo@uninsubria.it (A.C.); domenico.cavallo@uninsubria.it (D.M.C.); 2Center for Direct Reading and Sensor Technologies, National Institute for Occupational Safety and Health, Pittsburgh, PA 15236, USA; cuu5@cdc.gov; 3Centers for Disease Control and Prevention, Pittsburgh, PA 15236, USA

**Keywords:** low-cost sensors, miniaturized monitors, wearable monitors, implantable monitors, placeable monitors

## Abstract

Over the last decade, technological advancements have been made available and applied in a wide range of applications in several work fields, ranging from personal to industrial enforcements. One of the emerging issues concerns occupational safety and health in the Fourth Industrial Revolution and, in more detail, it deals with how industrial hygienists could improve the risk-assessment process. A possible way to achieve these aims is the adoption of new exposure-monitoring tools. In this study, a systematic review of the up-to-date scientific literature has been performed to identify and discuss the most-used sensors that could be useful for occupational risk assessment, with the intent of highlighting their pros and cons. A total of 40 papers have been included in this manuscript. The results show that sensors able to investigate airborne pollutants (i.e., gaseous pollutants and particulate matter), environmental conditions, physical agents, and workers’ postures could be usefully adopted in the risk-assessment process, since they could report significant data without significantly interfering with the job activities of the investigated subjects. To date, there are only few “next-generation” monitors and sensors (NGMSs) that could be effectively used on the workplace to preserve human health. Due to this fact, the development and the validation of new NGMSs will be crucial in the upcoming years, to adopt these technologies in occupational-risk assessment.

## 1. Introduction

### 1.1. Background

Since personal samplers were implemented in the 1960s, personal sampling has become a widely accepted practice (or, rather, the reference method) for exposure assessment in occupational hygiene [1,2]. Traditionally, personal sampling depends on relatively slow turnarounds between sample collection and subsequent laboratory analysis, which uses standardized methods to generate results, and this can limit the optimal implementation of workplace-risk-mitigation strategies in terms of promptness and efficacy [2,3]. On the other hand, “real-time” monitoring (i.e., by using “direct-reading” devices) allows for: (i) sensing the presence of specific hazards, (ii) collecting a data sample with a high temporal resolution, and (iii) having real-time feedback which, on the contrary, should be delayed when using the traditional approach of sample collection, followed by a subsequent off-line analysis [2,3,4]. These advantages, in occupational-hygiene applications, can potentially provide data of different nature for risk-management purposes; the information on the exposure to the hazard can be available in a more timely way, thus, the implementation of risk mitigation measures may be faster and more efficient (e.g., workers who receive real-time information can mitigate their own exposure, by changing their behavior and/or the procedure they are performing) [4]. The advent of wearable and unobtrusive sensors has allowed for measuring the parameters of interest (e.g., gas/vapor and aerosol concentrations, noise intensity, fatigue and heat stress) during real-life activities [5]. Further, data from real-time monitoring may have completely different characteristics (e.g., high volume of data, elevated generation velocity, heterogeneity of data), compared to those from traditional monitoring, which determines a great interest and challenge in the collection, storage, and modeling of data [6]. The development of cheaper (and, sometimes, also lightweight, miniaturized, and technically advanced) sensors (e.g., low-cost sensors—LCSs) and monitors (e.g., low-cost monitors—LCMs) is essential to promote the abovementioned activities and might have the potential to transform exposure-assessment approaches in occupational settings [7]. Following a definition already adopted in a previous publication [8], low-cost sensors and monitors are intended to mean that the cost of a single unit does not exceed the order of magnitude of a few hundred USD. The spread of a “next-generation” of sensor devices for occupational hygiene (i.e., low-cost, miniaturized, placeable, wearable, and implantable sensor technologies) and their role in the future of workplace-exposure assessment and risk assessment have recently been discussed [3,4,6,8]. Hereafter, a systematic collection of the up-to-date literature, aiming to outline the relative core topics, will be discussed.

### 1.2. Problem Statement

NGMSs (“next-generation” monitors and sensors), which refers to “miniaturized” and/or “wearable” sensors and/or monitors, are expected to make the exposure and risk assessment in occupational settings more convenient and comprehensive [4]. A recent systematic review [8] analyzed the use of NGMSs in occupational hygiene for airborne hazards: results outlined that these applications are less frequent than in environmental hygiene, and this is probably due to the fact that policy- and legislation-based decisions requires a high-level detection limit for data, precision, accuracy, and completeness [9]. Despite that, NGMSs devices can provide new resources in the occupational-safety and health-management fields [10,11,12,13,14,15,16,17,18,19,20,21,22,23]. Indeed, the studies considered in the above-mentioned systematic review, demonstrated overall that NGMSs provide useful data, if properly calibrated. NGMSs can also be easily adopted to improve exposure assessment studies in terms of spatio-temporal resolution, wearability (=prolonged use), and adaptability to different types of experimental designs and applications. For example, wearable sensors and devices could be used in various application fields such as: (i) ergonomic analysis [24], (ii) assessment of the weather’s effect on outdoor workers [25], or (iii) exposure to chemical substances which can affect workers’ health [26]. Personal-level sensors are also creating new opportunities for exposure-assessment studies. Technologies to study the environment, such as monitors and sensors, have always had a higher, more important role in the investigation of the occupational-risk-assessment process [27]. The application of Internet of Things (IoT) technologies also allows to connect NGMSs to each other and/or to smartphone apps and to upload data onto cloud platforms using Bluetooth or Wi-Fi technologies, which report back, in real-time, the recorded data [19,28]. Nevertheless, some drawbacks must be also considered. For example, NGMSs cannot be applied as reference-grade instrumentation in monitoring exposure to airborne chemicals for regulatory purposes [8]. NGMSs should be used paired with traditional methods for a period to allow the hygienist to calibrate the sensors in the most efficient way to obtain significant data for the risk assessment [29,30,31]. Another relevant issue regards power supply, in fact, most NGMSs cannot run for many hours without recharging [8]. Further, concerning ethics, a balance should be found among the respect for privacy and the intrusiveness that accompanies ubiquitous worker monitoring; mutual worker–employer trust must be achieved regarding the management of the large amount of data that can be generated by monitoring with NGMSs [4]. From a much more operational point of view, another weakness of NGMSs could be due to their misplacement during the workers’ activities. In fact, these situations may generate a lack of data quality, and it might be a serious problem for the industrial hygienist’s evaluations, in the case that the data should be analyzed for risk-assessment purposes [8].

### 1.3. Aim of the Study

The main purpose of this study is to analyze the scientific literature in order to understand the state-of-the-art technology concerning the use of NGMSs in the exposure and risk-assessment processes in occupational settings. In particular, the present study aims to focus on how NGMSs could be used in the field and on their practicability in the risk-assessment procedure. In addition, the authors decided to study several areas of interest regarding the technologies and the main technical aspects of wearable sensors in the field of industrial hygiene.

## 2. Materials and Methods

A rapid [32,33] systematic review of the literature was performed using the outcomes from Scopus database following PRISMA guidelines [34]. The main topic of interest involved in this work was about placeable, wearable, and implantable sensors and their application in occupational exposure assessment studies. Only scientific papers written in English were considered. A list of keywords was arranged in queries following the writing rules required by Scopus database, obtaining the final query (Table 1) adopted to get the papers from the database.

At the end of the research process, 40 papers were found in Scopus (Appendix A). The last research was conducted on 22 March 2022 (first research: 27 May 2021; weekly updates were performed starting in March 2022 until the submission date).

After reading each retrieved paper, the obtained information was schematically organized in a dedicated database by two of the authors (A.B. and G.F.). The papers were analyzed to find out information regarding the aims of the studies, the investigated risk factors, the application of NGMSs, and their technical features (e.g., sensor technology, devices’ dimensions, weight and cost, batteries performance, availability of mobile apps, connection and/or IoT technology, application (in the laboratory or in the field), and their availability (prototype or device on the market)). In the opinion of the authors, these are the most important topics regarding the implementation of the risk-assessment process in real, occupational settings. It should be noted that conducting a systematic review of the available evidence on the use of NGMSs in the exposure and risk-assessment process in occupational settings would have been a useful tool. It would allow the interpretation of the results of individual studies within the context of the totality of evidence and provide the evidence-base for guidelines or policy briefs. However, due to the high level of methodological rigor, systematic reviews require considerable time and skills to execute. When timely access to information is needed, “rapid reviews” instead of systematic reviews are a possibility that can be considered [32]. Rapid reviews are a form of knowledge synthesis, in which components of the systematic review process are simplified or omitted to produce information in a timely manner [33]. The present review is, therefore, configured as a rapid review, and the limitations that characterize this type of approach must be considered [32,35].

## 3. Results and Discussion

The review’s outcomes indicate that the number of research articles that involve low-cost sensors for the implementation of the risk-assessment process has steadily increased in the past decade, and it is constantly growing. Indeed, most (*n* = 38) of the scientific papers retrieved by applying the abovementioned research query were published from 2010 to 2021, and only two papers [36,37] were published before 2010. Seven studies [38,39,40,41,42,43,44] were performed in both actual-occupational and laboratory settings and seventeen studies [5,11,18,19,25,26,37,45,46,47,48,49,50,51,52,53,54] were conducted in applied field-based settings, while eight [31,36,55,56,57,58,59,60] were performed in a laboratory environment. At the time of writing (February 2022), most of the sensors used in the reviewed studies were available on the market. In fact, only 4 [43,48,57,59] out of 40 studies presented specifically designed prototype sensors and devices. Due to the ability of the involved sensors to detect various parameters (Table 2), only 11 [18,31,37,38,39,41,44,48,51,52,54] out of 40 studies focused on a single parameter.

### 3.1. Airborne Pollutants

#### 3.1.1. Gaseous Pollutants

Gaseous pollutants such as carbon monoxide (CO), nitrogen oxides (NOx), and ozone (O_3_) cause a range of deleterious respiratory- and cardiovascular-health effects [63]. The main airborne gaseous pollutants studied in the literature, using NGMSs technologies, are: (i) CO, (ii) oxidizing gases such as O_3_ and nitrogen dioxide (NO_2_), (iii) methane (CH_4_), (iv) sulfur dioxide (SO_2_), and (v) benzene (C_6_H_6_) (Table 2). In more detail, two papers [57,58] present new tools and techniques for the monitoring of airborne pollutants, in particular for C_6_H_6_, O_3_, and SO_2_. Leghrib and co-workers [58] used an array of plasma-treated metal-decorated carbon nanotubes for the quantification of airborne benzene, suggesting use for a selective, low-cost, and wearable sensor. Wan and co-workers [57] introduced a new miniaturized planar electrochemical-gas sensor for a rapid monitoring of multiple inorganic gases (i.e., oxygen (O_2_), O_3_, SO_2_, and CH_4_). In more detail, they presented the whole sensor’s construction process, outlining how, from a sensor available on the market, one could build a new customized, low-cost, and wearable device with good reliability to be applied for exposure and risk-assessment procedures [57]. The gas sensor consists of a porous polytetrafluoroethylene substrate, which allows fast gas diffusion and room-temperature ionic liquid as an electrolyte. To enhance adhesion between the electrodes and the substrate, for platinum-electrodes production, a metal-sputtering technique was used. Thus, compared with other already adopted gas sensors, the one proposed by Wan and collaborators is among the most promising toward a miniaturized, inexpensive, rapid-response, low-power, and multi-gas sensing system for the exposure monitoring of gaseous hazards [57]. Another work concerning the evaluation of CH_4_ concentrations was conducted by Shamasunder and co-workers [52], who tested the capacity of low-cost sensors for localized-exposure estimates. Johannessen and co-workers [38] presented a CO sensor’s fabrication process and the design of an IoT network to collect the real-time information gained by the instrument. In more detail, they designed and built a CO-sensor module employing a low-cost (i.e., USD 100) sensor that is commercially available. CO-sensor modules were built based on the EE-02 sensor (Exploratory Engineering, Telenor Digital AS, Trondheim, Norway), which enables long-range connectivity. After some tests conducted both in the laboratory and in the field at an incineration plant, they acknowledged that the best choice to detect rapid and short-term variations of CO levels in workplaces could be real-time monitoring conducted for an extended period of time. However, the investigation of the sensor accuracy and resistance toward interfering gases, is and will be, a crucial point in the evaluation of CO-occupational exposure. Regarding CO_2_, its monitoring could be developed as a new approach to alert individuals using all forms of respiratory support, when breathing becomes stressed before overt symptoms appear. For example, Pleil and co-workers [48] have analyzed results from recent experiments employing in-mask (Tunable Laser Spectroscopy-TLS) CO_2_ sensors, to evaluate if these could become a reliable early-warning tool. Assessing the accuracy of the commercially available sensors is an important step to understand their application in specific tasks, where future research should be focused on. In this regard, Isiugo and co-workers [26] evaluated the performance of different gas sensors for the measurement of ambient O_3_ and NO_2_. Results of their work showed that the performance of the tested instruments was influenced by environmental conditions and that, overall, only one of the three tested sensors had an appropriate accuracy. Zuidema and colleagues [11,45] realized a multi-sensor network using low-cost sensors distributed in the work environment. The multi-sensor network was used to design hazard maps of a heavy-vehicle-manufacturing facility. This could offer insight into the sources, areas of high/variability in concentrations near different activities, and distribution of hazards. The most relevant issue emerged from this study was related to the need of a proper calibration of the instruments. In fact, all the sensors were primarily calibrated in laboratory and then underwent field calibration. The multi-hazard network acquired data continuously for 5 months: all the data gained by the sensors were collected in a database and then used to obtain a hazard map of the facility for each investigated airborne pollutant (i.e., CO, PM, O_3_, and NO_2_). The so-obtained hazard maps could be used to evaluate if the adopted control strategies are effective, offering an advantage over traditional industrial-hygiene approaches. Further, the mapping tools provided by a multi-hazard sensor network, combined with information on workers’ locations, can be used to estimate personal exposure to multiple occupational hazards. For this purpose, personal direct-reading instruments were deployed for the same contaminants, to evaluate the ability of the multi-sensor network to potentially provide personal-exposure estimates for any employee whose position can be tracked. Available studies also outline other benefits of a multi-sensor network and/or personal instrument, based on low-cost, customizable, low-power-demand sensors. In fact, this instrument configuration could be useful to identify and divide the working areas by their levels of hazardousness, helping all the workers to control their level of exposure to hazardous pollutants, according to the respective occupational-exposure-limit values [14]. The most relevant problem rising from the available literature is that the measurement error derived from low-cost sensors can often be attributed to issues of sensitivity and specificity, in part due to sensor drift, degradation over time, or responsiveness to non-target species. This problem must be further investigated, especially in terms of instrument accuracy, which must be improved, despite the fact some laboratory results for particular pollutants are very promising [57,58]. A characteristic that arises from the reviewed studies is the customizability, intended as the propensity of the monitors to be easily adapted, integrated, and assembled with other components, which can be developed starting from a sensor already available on the market.

#### 3.1.2. Particulate Matter

Epidemiological and toxicological studies show that a number of negative effects on human health are possibly related to particulate matter (PM) exposure [64]. Recently, miniaturized, low-cost sensors for PM have become increasingly available, making possible a sensor network to elaborate and characterize maps of particle concentration with high spatial and temporal resolution [45]. Among the reviewed papers, three different on-field studies [45,47,54] that used low-cost and wearable sensors for PM measurements were available. These were performed in three different occupational settings, namely a heavy-vehicle-manufacturing facility, an agricultural setting, and hairdressing salons. The first study [45] aimed to design a method that uses hazard-mapping data to optimize the number and location of sensors within a network for a long-term assessment of occupational-PM concentration. The proposed protocol is based on a statistical methodology to define the eventual removal order of the sensors located in a manufacturing facility, to determine their optimal location based on preliminary hazard-mapping data. The main aim was to preserve the locations with higher temporal PM variability, to produce the most accurate hazard maps. The statistical methodology presented by Berman and colleagues [46] is very promising because, although in this case it is used for the analysis of PM, it could be modified and used for different hazards and occupational settings to obtain very accurate hazard maps and to implement risk assessment. For example, this methodology could be helpful for a large-scale preliminary-monitoring campaign. Then, based on the obtained results, a subsequent monitoring campaign could be planned based on a reduced number of optimally placed sensors, to perform long-term-exposure assessment. Another study by Zuidema and co-workers [41] is somehow connected to the abovementioned study, since it presented an example of how to apply a correction factor to low-cost sensors to obtain the most-possible-performant sensor network in a heavy-vehicle-manufacturing plant. In the second study [54], three different sensors were used to assess the PM exposure of farmworkers. The first two sensors involved in the study were low-cost sensors (i.e., OPC-N3 by Alphasense Ltd., Essex, UK, and AirBeam2 by HabitatMap, Brooklyn, NY, USA, which is composed of the PMS7003 PM sensor produced by Plantower) and the third was a higher-cost device (i.e., GRIMM Mini-WRAS 1371 by Grimm Aerosol Technik GmBH & Co. KG, Ainring, Germany). The study was conducted over 5 non-consecutive days, from 8:00 a.m. to 4:00 p.m. Results outlined that OPC-N3 performed better than AirBeam2 and if compared to a gravimetrical filter measurement, had generally higher averages than the filter concentration. However, after excluding the data point where the air-sampling pump failed to run, they found a good agreement between OPC-N3 and the filter measurement. These findings suggest that OPC-N3 may be suitable for some agricultural-exposure measurements. The article by Shao and colleagues [47] was related to a pilot study concerning the exposure assessment of PM among hairdressers. Indoor PM concentrations in hair salon were characterized, and the performance of three low-cost sensors (uRAD A3 by Winsen ZH03A PM sensor Magnasci SRL, Romania; Flow by Plume labs; AirVisua Pro, AVPM25b PM sensor, USA) were compared to a portable monitor (DustTrak, 8530, TSI Incorporated). The results of the tested low-cost sensors were very promising. Among these, the uRAD and AirVisual were the sensors that tracked better with their reference device (DustTrak) during most of the sampling time. Contrarily, the FLOW low-cost sensor did not perform as well. Overall, these studies outlined that several NGMSs characterized by acceptable performance are available, but not all of them perform as well as the more expensive instruments taken as reference in these evaluation studies. For this reason, to date, NGMSs cannot totally replace traditional approaches in occupational-exposure assessment, but they can fill other gaps, such as improving data in terms of spatial and temporal resolution [8].

### 3.2. Other Risk Factors for Workers’ Health and Safety

The results of the studies included in this review outlined that new sensor technologies can be used for the evaluation of other relevant workplace-related health and safety risk factors. These include: (i) physical agents and worker’s physiological parameters related to thermal stress/strain and (ii) posture assessment (to prevent work-related musculoskeletal disorders—WMSDs).

#### 3.2.1. Physical Agents and Workers’ Physiological Parameters

##### Temperature

Physiological-temperature monitoring could improve both safety monitoring and work–rest planning, to maximize effective and safe performance (a safe work environment is where workers are physically capable of doing all the required job tasks [19]). There are two key research problems in monitoring thermal-work stress: (i) accurately determining an individual’s thermal-work strain and (ii) using the thermal-work-strain state to optimize human performance [65]. Among those reviewed, two studies were performed in cold environments [19,53], and four studies focused on hot environments [5,25,36,65]. Most of these studies used the same instrumentation (i.e., Thermochron iButton, HOBO Pendant Temperature, and Garmin vivoActive HR) to measure physiological metrics (i.e., body temperature and heart rate—HR) which were employed to calculate the thermal stress and strain of the investigated subjects. All these devices are wearable tools available for the consumer on the market. The objective of the study conducted by Sugg and co-workers [53] was to assess the personal ambient temperature (PAT), which is intended as the personally experienced ambient temperature among workers in a cold environment. The symptoms of prevailing chronic diseases are aggravated by cold weather. Scarce physical and mental performances, due to uncomfortable thermal sensations, numbness of hands, and lower body temperatures, could be influenced by low temperature [53]. In this study, workers reported on their experience concerning several cold-related issues, ranging from numbness in the hands and feet to shivering. Especially for outdoor-PAT data, results outlined that the ambient temperature information coming from personal monitoring devices generally agreed with ambient-weather-station data. Significant differences among devices, despite usage by the same participants, was highlighted. The authors reported a great variability among the same subjects, depending on the sensor choice and placement. This is an issue that affects the identification of participants who wore their devices improperly. The long reaction times of the iButton and HOBO devices to temperature measurements may be a limitation of small-scale spatiotemporal studies, because workers could move between different microenvironments in a short period. Further information on this topic is presented by Nelson and colleagues [19]. The authors also focused on occupational workers in cold environments using Thermocron iButtons to collect ambient temperature, but the aim of their study was to understand the workers’ response to (i) the report-back process, (ii) their perception of exposure to cold environments, (iii) the potential of behavioral modification, and (iv) the understanding of personal-biomonitoring results (i.e., heart rate). A report-back packet that displayed the study’s results, containing biomonitoring information, was given to participants at the end of data collection. After that, a survey to assess potential behavioral modifications and preferences of health-data formatting was conducted. Participants found this process very useful; in fact, they expressed a greater willingness to modify their occupational behaviors to reduce their cold exposure. In terms of promoting behavioral change to cold temperature, the results of this study suggest that reporting the outcomes of each worker could be an effective way to protect workers from these problematics. Not only is the exposure to cold environment is a factor of hazard, but also occupational heat exposure is a crucial workplace hazard, and it is related to increases in health-related illness and injuries due to fatigue, as well as declines in safety and worker vigilance. Workers’ performance and productivity can also be affected by exposure to heat, which may cause their decline [53]. In recent years, there were several improvements in low-cost, wearable-sensor technology, aiming to study the individual’s daily temperature exposure using single-point measures, which provide an up-to-date method to improve the scale of detailed time-location records and important knowledge about microclimate variability for outdoor workers. Sugg and co-workers [25] conducted a study to demonstrate space and time patterns of PAT exposure and feasibility of using wearable sensors to measure PAT in an at-risk group of outdoor workers. In more detail, the authors aimed to: (i) show the ability to evaluate PAT exposure in both space and time, (ii) characterize site-specific and personal variability in PAT, and (iii) examine how PAT varied between multiple microenvironments. To do that, each participant was equipped with Thermochron iButton devices positioned outside of their collar with the devices facing outward, to collect data regarding ambient environmental conditions. Moreover, a few participants wore Garmin Vivoactive HR watches, to acquire locational and contextual data on heart rate. Sugg and co-authors found out that indoor workers, compared to outdoor ones, have a wider choice about where they spend their workday and may have more opportunities and resources to mitigate their exposure. Despite that, nearly 90% of the participants found the information provided to them useful in mitigating their own heat exposure while at work. A particular study was performed in laboratory setting on firefighters by Coca et al. [36]. The authors tried to demonstrate the accuracy of wearable sensor in this type of working activity. Indeed, firefighters experience tremendous physical stresses in the course of their duties, both metabolically and environmentally. In this study, plethysmographic sensors (LifeShirt System, VivoMetrics, Ventura, CA, USA) incorporated into a vest were used. Some of the physiologically variables monitored by these wearable devices were heart rate, respiratory rate, skin temperature, oxygen saturation, tidal volume, and minute ventilation. Physiological data were stored onto a small, portable data recorder carried in a pouch attached to the vest and telemetered in real-time to a laptop computer. The conclusion of this study indicates that LifeShirt Systems measure somewhat accurately within the hot, moist environment of standard firefighter gear. In fact, most of the physiological outcomes were not statistically different from the physiological data recorded on standard laboratory equipment (i.e., 12-lead ECG skin electrodes, skin-temperature sensor (SQ2020-1F8, Grant Instruments Ltd., Cambridgeshire, UK), Nonin X-pod pulse oximeter (Nonin Medical Inc., Plymouth, MN, USA)). This study obtained very promising results, but it also suggested that additional experiments in actual firefighting scenarios are warranted to determine the accuracy in field settings.

##### UV Radiation

In addition to temperature exposure, the exposure to ultraviolet radiation (UV) is also a relevant issue, especially for outdoor workers. Long exposure to the sun results in an over assumption of UV rays, which has a beneficial effect on human physiology in a normal dose, but overexposure could lead to several diseases, such as DNA mutations and subsequent skin-cell carcinoma, benign tumors, fine and coarse wrinkles, mottled pigmentation, and other cellular-proliferative diseases [66]. To avoid all these potential risks, the workers must be provided with alerts about overexposure events. Pievanelli and colleagues [59], in a conference paper, presented an operational scheme for the realization of compact, wireless sensors that are able to detect physical agent (i.e., UV) exposure, which are suitable for the protection of workers employed outdoors. The platform is made up of mobile, wireless sensor nodes, designed to be attached on clothes, to be simply worn by anybody. Even if the whole set of components has been identified and tested, the single elements have not been integrated together. Further, among those reviewed, we found two manuscripts regarding in-the-field UV exposure [37,61]. Sabburg et al. [37] collected UV-A irradiance data to quantify the effect of clouds in UV-A exposure, using an integrated sky camera and radiation system during an autumn and winter period. Baczynska et al. [61] over 2016 and 2017 measured the in-flight UV exposure of pilots in England, using GENESIS-UV sensors for measurement inside the cockpits. Commercial pilots are a particular category of workers, due to the fact that they are at twice the risk of melanoma and skin cancer than the general population [67,68]. Pilots are exposed to solar and UV radiations that may be significantly higher at flight altitudes than on the ground. In this study, sensors were clipped to the shirt at chest levels, and the pilots had also to fill a diary that included date, time, and other flight information. From this study emerged that the direct method of in-flight spectral measurement is challenging, and the use of small, wearable sensors may be a promising solution. However, wearable sensors often cannot be used for measurement of the solar radiation that is filtered through aircraft windshields, without correction factor. As a matter of fact, in this study, the GENESIS-UV sensors had strong wavelength-dependence and needed a correction factor to make the acquired data evaluable.

##### Noise

Among those reviewed, three papers focused on occupational exposure to noise. Two of these were performed by Zuidema and co-workers [11,45] and in each one a custom sound-pressure-level (SPL) sensor was used to assess the workers’ exposure to noise in a heavy-vehicle-manufacturing facility. Results obtained from the custom sensor were compared to those obtained by means of a reference instrument, which was the model “XL2” (NTi Audio AG, Liechtenstein). The noise sensor developed for these two studies is composed of a microprocessor that is plugged into an omnidirectional condenser microphone, and it was calibrated by playing a calibration sound, with an acoustic generator and an amplifier, between 65 and 95 dB. The sensor’s response was then compared to the reference sound level. Since the noise is a physical hazard that does not disperse from a certain source the same as other pollutants, the sensors’ network that had been created had a limited ability, due to the abovementioned issues, to capture impact or impulse noise. Misistia and colleagues [40] conducted a study to assess the response to the BOP (blast over pressure) of wearable sensors against industry-standard pressure transducers. In this case, the Tourmaline ICP pressure-sensor model (PCB piezometric, Depew, NY, USA) was adopted to evaluate the sensor’s orientation error in military-personnel helmets. The experimental procedures were conducted under controlled laboratory conditions using a shock tube, and some of the findings were verified in the field. The sensor used for this study was the Black Box Biometrics (B3) Blast Gauge. On this sensor, a scheme is printed to identify the specific location where it must be worn, namely on the back of the helmet, on the left shoulder, and on the chest. The authors individuated three factors that might influence the recorded-pressure values from wearable sensors, which are: (i) the orientation of the body in respect to the sources of the blast waves, (ii) the intensity of the shock waves, and (iii) the local geometry of the ambient around it. Results of this study revealed that there is an underestimation error in the reflected pressure for B3 sensors, but the incident overpressure peak is comparable to the PCB, so the impulse values are overestimated, regardless of the tested configuration of the B3s. Despite Misistia’s study focused on military personnel, the results could be extended to all workers exposed to high-frequency noise. The B3s’ configuration could be easily worn during the work shift, and it might report back interesting data to compare to the standard PCB sensors present in the workplace, thanks to the wireless communications capabilities of the instrument.

##### Laser

Concerning hand-held Laser exposure, from an occupational-risk–management point of view, the market launch of hand-held laser-processing devices should be closely related to the safety of the machines. Personal protective equipment such as protective eyewear or clothing must not be considered, as in any risk-management procedure, as the first choice to prevent injuries and manage the risks in workplaces. Indeed, these strategies are adopted only in those cases when it is not possible to eliminate the sources of risk. In our literature research, we found a paper regarding this problem that was performed in a laboratory setting by Puester and colleagues [18,51]. This study aimed at the qualification and adoption of safety measures for the use of a hand-held laser instrument. The sensors selected to conduct the laboratory investigation were: (i) tactile sensors, (ii) inductive sensors, (iii) capacitive sensors, (iv) ultrasonic sensors, (v) inclination sensors, (vi) acceleration sensors, (vii) gyroscopes, and (viii) temperature sensors. Depending on the output and on the body parts’ distance to the process zone, if the laser radiation become accessible, critical irradiance on the human body can occur. Apart from irradiance, exposure time is the second critical value. To avoid lesions, laser radiation must be isolated or deactivated as soon as possible under fault conditions. The investigations reveal solutions to equip laser devices with safety-related parts and safety control to minimize the risks from laser radiation.

##### Mechanical Vibration

Mechanical vibrations are known to affect the hand–arm system or the whole body of workers who use machines or equipment that produce vibrations. Austad et al. [56] used the IsenseU, a flexible, wearable, and robust sensor, suitable for being integrated into clothing, to assess the hand–arm-vibration exposure in a laboratory setting. The findings of this study showed that the IsenseU sensor can be useful for estimating the vibration-exposure time, the frequency-weighted acceleration, and daily-exposure values. Moreover, if the sensor might be integrated into the sleeve of a jacket, vibration-exposure measurement could be performed concurrently with skin- and ambient-temperature measurement. To conclude, in our review of the literature, we found that there are relatively few studies focused on next-generation sensors for assessing exposure to physical agents. This topic should be further investigated to obtain more complete information to improve risk-assessment processes in workplaces. The most tempting prospect is that next-generation work instrumentations and personal protective equipment should be equipped with new, small sensors that could provide real-time feedback about emissions and/or the exposure of workers to work-related risks.

#### 3.2.2. Posture Assessment and Work-Related Musculoskeletal Disorders

MSDs (musculoskeletal disorders) can be defined as a group of disorders or injuries that could deform a subject’s inner body while it is stressed. Examples of MSDs include bursitis, carpal tunnel syndrome, and tendonitis [24]. Work-related musculoskeletal disorders (WMSDs) refer to MSDs that are due to workplace activities associated with physical job tasks. According to the Occupational Safety and Health Administration (OSHA), which is based in the United States, there are eight risk factors related to WMSDs, including (i) extreme temperature, (ii) repetition, (iii) static postures, (iv) vibration, (v) quick motion, (vi) compressed or contact stress, (vii) force, and (viii) awkward postures [69,70]. Most of the time, awkward postures can be prevented by re-setting the workplace layout or selecting a proper tool for workers, but different work tasks are affected with different types of risks, so the challenge is to find out new customized solutions that can solve the specific issue. A specific job-hazards analysis could identify the workplace’s risk, but it may be tricking to carry it out because of the complexity of the job and the manual effort needed to monitor work processes [71]. Among those reviewed, five different papers were found regarding this topic [24,42,43,49,62], and the main outcomes are reported here. In recent years, wearable sensors have been used for quantitative instrumental-based biomechanical-risk-assessment studies to prevent work-related musculoskeletal disorders. Instrumentation-based tools are generally not included in the standardized methods for biomechanical-risk-assessment studies. because the ones commonly used are based on observational and subjective approaches. The spread of Industry 4.0 may represent a new scenario in which the computational capabilities and network connections that characterize smart, wearable sensors are able to be transparent, sensitive, responsive, and adaptive to workers’ movements, allowing for real-time, online monitoring of working tasks. Recently, several methods have been developed, accepted by the international literature, and used in the workplace to attempt to reduce the WMSDs. About this, the most innovative wearable technologies and the electronic smart devices that support these types of investigations to improve the biomechanical-risk assessment, adapt them to all the work situations and outline the limits of the up-to-date standardized methods, without interfering with the workers’ activities. This allows real-time estimation of the risk, providing direct feedback to the end-user, who is constantly monitored directly while at work. Several commercial, wearable inertial sensors have the possibility to stream data to a remote computer or a web server in real-time. This allows for recording, processing, and reviewing sensor data online and affording new opportunities in near-real-time, for rapid feedback about work postures to subjects or to managers and supervisors. Moreover, body-worn inertial sensor technology provides several opportunities to improve the safety and health of workers who do physical tasks [62]. Despite the widespread use of these new tools, there are still too few scientists and engineers predicting the use of wearable technologies for biomechanical-risk assessment, although (i) the need to obtain increasingly quantitative evaluation, (ii) the recent miniaturization process, and (iii) the need to stay updated with a constantly evolving manual handling scenario are asking for their use. Therefore, regarding biomechanical-risk assessment, the adoption of new innovative technologies is at an initial stage [42]. Concerning MSDs, construction jobs are one of the most labor-demanding compared to other industries. Often, construction workers exceed their natural physical capability to make up for the increasing challenges and complexity in this business. Due to this fact, construction jobs are among the most ergonomically hazardous, because they often involve activities such as body twisting, manual handling, heavy lifting, and working in awkward positions, which are all potential causes of WMSDs in workers. The most common ones are tendonitis, sprains, back pain, strains, and CTS. The postures of different body parts re generally measured in terms of the degree of bend from the neutral posture, to identify the risks associated with the awkward postures. Sensor-based direct measurement of risk factors provides a great opportunity for unobtrusive and precise ergonomic assessment of construction tasks. Nevertheless, calibrating, setting up, and using a complicated sensor network requires expertise that is normally less than what is expected from most construction workers and field workers. Even if such technologies are on the market, the economical effort, as well as the time commitment necessary to purchase, install, and maintain the tools, may be considered an impeding factor. Commonly, the most reliable sensor used for biomechanical-risk assessment is the IMU (inertial measurement unit) [42]. These sensors allow for the measure of the orientation, position, velocity, and acceleration of the body posture. An important study on selected subjects was made by Nath and co-workers [24] in a laboratory environment. The authors used a “two smartphones” configuration, through the devices’ 3D accelerometer sensor, to demonstrate the potential of mobile devices in ergonomic assessment. For data treatment, they used the sensory ones gained by the smartphones, which were mounted on the worker’s upper-arm and waist, while the worker is performing a task. The posture during a screwdriving task was analyzed, and, in this case, the position of the two smartphones produced the most distinctive features for most manual jobs performed by field workers. The data collected from the smartphone on the upper arm were used for measuring total flexion, while the data collected by the smartphone mounted on the waist were used to measure trunk flexion. Accelerometers in IMUs have a generally higher sensitivity than those in smartphones, but for static postures, a smartphone’s built-in inertial sensors are as reliable as other standard tools, since they are commonly equipped with a high number of sensors that can be activated to collect several types of data. This type of accelerometer seems to report back significant and useful data to assess any anomalies in the body postures. The results presented focused on posture analysis for trunk and shoulder flexions, but, with a few modifications for other types of field activities (e.g., manual tasks, manual handling, and manual lifting), the developed methodology and the analysis techniques can be generalized. Moreover, the proposed method is applicable for various occupations that are exposed to WMSDs due to awkward positions.

## 4. Conclusions

The outcomes of this review indicate that the number of research articles involving NGMSs for the implementation of the risk-assessment process has steadily increased in the past decade, and it is constantly growing. With the spread of the Fourth Industrial Revolution (“Industry 4.0”), the main problem of industrial hygiene (“occupational hygiene 4.0”) is how to improve the risk-assessment process in these new high-tech plants, while updating the traditionally adopted procedures. The concept of Industry 4.0 is the implementation of industry toward an intelligent model, in which collaborative robotics and new technologies interconnect workers and machine tools [6]. To properly preserve workers’ safety and health during (and beyond) the Fourth Industrial Revolution, in the last 10 years, the interest in wearable and low-cost sensors has been increasingly growing. The state-of-the-art technology regarding wearable sensors is in the early stages, as it is mostly considering some specific work-related health and safety parameters. To date, there are only few NGMSs that could be properly used at a workplace for exposure-assessment and risk-assessment purposes. Despite that, due to the continuous advancement in new technologies, the performance and the number of NGMSs will be further improved, always obtaining more advanced sensors. Smart devices currently available on the market must be considered as a resource. An advantage in the usage of small, wearable sensors is the possibility to obtain a complete dataset over the entire work shift, even though the reliability of the batteries of these new sensors must be further improved and designed. Moreover, it was highlighted that significant differences among devices could occur, despite their usage in the same conditions. Additionally, a fundamental step that is mandatory before the usage of these technologies is the evaluation of the performance and the reliability of the sensors. To conclude, a preliminary study about the new technologies has been conducted. The authors are also confident that the scientific research regarding these topics could improve the up-to-date available literature, to support the development of proper instrumentations to implement the risk-assessment process.

## Figures and Tables

**Table 1 sensors-22-04841-t001:** Query used in the Scopus database.

Database	Search Query
Scopus	(TITLE-ABS-KEY (“sensor*” AND “occupation*”)) AND (TITLE-ABS-KEY (“occupational exposure” OR “human exposure” OR “exposome” OR “miniaturized sens*”)) AND (TITLE-ABS-KEY (“sensor network” OR “wearable sens*” OR “crowd sensing” OR „participatory sensing” OR “mobile sensor node” OR “low cost sensor” OR “citizen science” OR “mobile phone app*” OR “lightweight device*” OR “bluetooth” OR “air pollution sens*” OR “portable device” OR server OR cloud OR “miniaturized sensor*”))

**Table 2 sensors-22-04841-t002:** Parameters investigated as occupational-risk factors, sensors used for measurement and assessment of these risk factors, and relative technologies.

Risk Factor	Sensor Name/Model	Sensor Technology	References
**Airborne Pollutants**			
Carbon Monoxide (CO)	IOT-CO-1000	EC	[38]
	Alphasense CO-B4	EC	[11,45]
	n.a.	MDCNTS	[58]
Carbon Dioxide (CO_2_)	n.a.	TLS	[48]
Nitrogen Dioxide (NO_2_)	Cairclip NO2	EC	[26]
	Alphasense OX-B431	EC	[11,45]
	n.a.	MDCNTS	[58]
Ozone (O_3_)	Cairclip O3	EC	[26]
	Alphasense OX-B431	EC	[11,45]
	MPRE	EC	[57]
Methane (CH_4_)	Figaro TGS 2600	MOS	[52]
	MPRE	EC	[57]
Benzene (C_6_H_6_)	n.a.	MDCNTS	[58]
Hydrogen Sulfide (H_2_S)	n.a.	MDCNTS	[58]
Sulphur Dioxide (SO_2_)	MPRE	EC	[57]
Particulate Matter (PM)	Sharp Electronics GP2Y1010AU0F	LS	[11,41,45,59]
	Alphasense OPC-N3	LS	[43,51]
	uRAD model A3	LS	[47]
	Plumbe labs. FLOW	LS	[47]
	AVPM25b AirVisual Pro	LS	[47]
	Sensirion SPS30	LS	[31]
	Airbeam2	LS	[31]
	Plantower PMSA003	LS	[31]
**Other Parameters**			
Temperature (T)	Thermocron iButton DS1921G	TH	[19,25,47]
	HOBO Pendant Temperature	TH	[19,53]
	Garmin vivoActive HR	n.a.	[19,53]
	LifeShirt	n.a.	[36]
	Vital Jacket	n.a.	[5]
	PT100	TH	[18,50]
	AM2302	TH	[11,45]
Ultraviolet Radiation (UV)	GENESIS-UV	n.a.	[61]
	n.a.	SiC	[59]
Noise	n.a.	SPL	[11,45]
	B3 (Black Box Biometrics Blast sensor)	SPL	[40]
Work-related Musculoskeletal Disorders (WMSDs)	IsenseU	IMU	[56]
	n.a.	IMU	[42,43]
	Hand-Heald dynamometer	IMS	[42]
	EMG	EMA	[42]
	n.a.	IMU	[24,62]
	G-Sensor	IMU	[49]
Proximity/Collision Accidents	AVM	RFID	[50]

EC—electrochemical sensor; MOS—metal-oxide semiconductor; MDCNTS—metal-decorated carbon-nanotube sensor; TLS—tunable-laser spectroscopy; LS—light scattering; TH—thermistor; SiC—silicon-carbide sensor; SPL—sound-pressure level; IMU—inertial-measurement unit; IMS—isometric-muscle strength; EMA—electrical-muscle activity; RFID—radio-frequency identification; n.a.—not applicable) and references to the papers in which sensors were made explicit and used.

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
