# Peer review of "Evolution and Applications of Recent Sensing Technology for Occupational Risk Assessment: A Rapid Review of the Literature"

_sensors, 2022, doi:10.3390/s22134841_

Round 1

Reviewer 1 Report

The article does not represent any value for specialists. The literature search is very limited. It did not touch upon publications that actually analyze advances in the development of sensors that allow you to control the state of the surrounding atmosphere and measure temperature, dust content, noise, vibrations, etc.

The discussion of the sensors is very superficial. This is not a review, but a simple listing of some of the sensors mentioned in some publications. This article is not aimed to analyze the advantages and disadvantages of sensors that can be used to assess occupational risks.

In addition, the authors forgot to include in the factors affecting working conditions, such as air humidity, illumination level, air velocity, and the level of electromagnetic radiation.

From my point of view, this manuscript is too a "rapid review".

Therefore, I cannot recommend this article for publication.

Author Response

Dear reviewer, thank you for your considerations which challenges us to improve our work. Nevertheless, regarding this manuscript and especially these new tools for the occupational risk assessment, the authors are convinced that it could be a good, although rapid, up-to-date state of the art of scientific literature. Then, we are aware that some limitations may afflict this manuscript: we added this sentence in the “Materials and Methods” section:

“It should be noted that conducing a systematic review of the available evidence on the use of NGMSs in the exposure and risk assessment process in occupational settings would have been a useful tool. It would allow the interpretation of the results of individual studies within the context of the totality of evidence and provide the evidence-base for guidelines or policy briefs. However, due to the high level of methodological rigor, systematic reviews require considerable time and skills to execute. When timely access to information is needed “rapid reviews” instead of systematic reviews is a possibility that can be considered [32]. Rapid reviews are a form of knowledge synthesis in which components of the systematic review process are simplified or omitted to produce information in a timely manner [33]. The present review is therefore configured as a rapid review and therefore the limitations that characterize this type of approach must be considered [32,35].”

Reviewer 2 Report

I appreciate authors reviewed recent sensing technology for air pollutant monitoring. I think that they have completed reviewed and discussed these literatures. I have only two comments:

  1. Please write the full name for “n.a.” under the footnote of Table 2.
  2. I suggest that authors should cite occupational epidemiological study, not environmental epidemiological study. Because this manuscript focuses on the occupational risk. For example, in lines 263 to 264, the review article from Dr. Brook focused on general environments, not occupational environments.

Author Response

  • Please write the full name for “n.a.” under the footnote of Table 2.

Dear reviewer, thank you for the suggestion but the full name for “n.a.” has already been reported in table 2 caption (now at line 197).

  • I suggest that authors should cite occupational epidemiological study, not environmental epidemiological study. Because this manuscript focuses on the occupational risk. For example, in lines 263 to 264, the review article from Dr. Brook focused on general environments, not occupational environments.

Dear reviewer, we appreciate your comment which contributes to improving the manuscript. Reference to Brook et al. was substituted citing Fang and coworkers (Fang, Cassidy and Christiani, 2010) who refer to Occupational Exposure to Particulate Matter.

Reviewer 3 Report

At line 100 the authors have higlighted some drawbacks of NGMSs. I'd like to add another one not mentioned: the battery duration do not allow intense measuring or certain types of sampling campains.... furthermore some chemical sensors require a warm up that makes impossible to put the monitoring device in low power mode. PM sensors often include a fan that forse air into the measuring chamber of the sensor in that case the fan is always on and energy consuming despite the low capacity battery used.  That means that battery duration will be a a great challenge in the field of wearable sensors.

Author Response

  • At line 100 the authors have higlighted some drawbacks of NGMSs. I'd like to add another one not mentioned: the battery duration do not allow intense measuring or certain types of sampling campains.... furthermore some chemical sensors require a warm up that makes impossible to put the monitoring device in low power mode. PM sensors often include a fan that forse air into the measuring chamber of the sensor in that case the fan is always on and energy consuming despite the low capacity battery used.  That means that battery duration will be a a great challenge in the field of wearable sensors.

Dear reviewer, thanks for your comment which is more than correct. We added a citation at line 101 as suggested. The authors totally agree with you, in fact we had already underlined this issue in the conclusions section (now at line 605-606).

Round 2

Reviewer 1 Report

Dear authors, to your regret, my opinion about your so-called "review" has not changed. I never found a synthesis of knowledge in your review. I understand that real reviews take time and skill to prepare. As I understand it, you do not have the necessary qualifications, and you do not want to waste time. I do not want other researchers to waste time reading your "review".